#### A preliminary study on the comprehensive threshold for debris-flow early warning

Xiaoqiang XUE, Jian HUANG State Key Laboratory of Geohazard Prevention and Geoenvironment Protection Chengdu University of Technology, Chengdu, Sichuan 610059, China E-mail: huangjian2010@gmail.com

# Abstract

Debris-flows not only cause a great loss of property, but also kill and injure people every rainy season in the mountainous regions of China. In order to reduce hazard and risk, several methods of assessing rainfall thresholds have been provided at present, based on statistical models. However, the limited rainfall data with debris-flow occurrence or non-occurrence makes threshold analyses very difficult. This paper, therefore, presented a kind of comprehensive threshold consisting of pore-water pressure from Terzaghi theory, and rainfall factors from frequent usage for predicting debris-flow occurrence. Rainfall and pore pressure data has been collected in a number of locations in Wenjiagou gully to assess critical rainfall and pore pressure values for debris flow initiation. The three-level early warning criteria (Zero, Attention, and Warning) has been adopted and the corresponding judgement conditions has been defined based on monitoring data in a real-time way. Finally, it is suggested that the combination of these two critical values might be a useful approach in a warning system for safeguarding of population in debris-flow prone areas.

Keywords: debris flow, comprehensive threshold, pore-water pressure, Wenjiagou gully

## 1. Introduction

A great number of debris-flows occurred in mountainous area every year in a rainy season. The fast population increase and high speed economic development in these areas always caused considerably catastrophic accidents and socio-economic losses. On Aug. 7 2010, a giant debris flow from Luojiayu gully and Sanyanyu gully at Zhougu County, Gansu Province, China, killed 1765 people living on the densely urbanized fan (Tang et al. 2011). Additionally in Southwestern China, the Wenchuan earthquake on May12, 2008, Yushu earthquake on April 14, 2010, Lushan earthquake on April 20, 2013, Ludian earthquake on August, 3 2014 and Nepal earthquake on April, 25 2015 trigged thousands of landslides and cracked mountains which are easy to develop into debris-flow under rainstorm condition. It's much similar to the Chi-Chi earthquake area (Taiwan), where numerous co-seismic landslides were triggered as well, and causing the continuous debris flows for 10 years after the earthquake (Yu et al. 2013b). These catastrophic events also indicated that the human vulnerability to natural hazards as well as the lack of knowledge on natural disaster prevention and mitigation. So that there is an urgent demand for an effective method to reduce the hazard and risk. Therefore, researchers have been working on forecasting debris-flow occurrences and setting up early warning systems. Especially on the regional scale, the methods for debris-flow early warning are frequently based on statistical models which have already been proved their importance in predicting debris-flow occurrence (Baum and Godt 2009; Guzzetti et al. 2007b; Keefer et al. 1987; Segoni et al. 2014; Shuin et al. 2012; Tropeano and Turconi 2004). Several parameters were selected for the assessment of rainfall thresholds mainly including rainfall intensity and duration (Cannon et al. 2008; Guzzetti et al. 2007a; Guzzetti et al. 2007b; Keefer et al. 1987), antecedent precipitation (Glade et al. 2000), and cumulative rainfall(Guo et al. 2013). Baum and Godt (2009) used a combination of a cumulative rainfall threshold, rainfall intensity-duration threshold and antecedent water index or soil wetness for shallow landslide forecasting.

Although widely used in the mountainous areas, these approaches are currently affected by some drawbacks which still restrain a fully operational application to early warning systems. One of the main problems is the lack of available data about rainfall with debris-flow occurrence or non-occurrence. Parameters selected for forecasting debris-flow occurrences are commonly limited to rainfall information, especially for a single gully. Chenyulan River Watershed in Taiwan, there are many debris flows triggered by Typhoon each time. After previous research identified 47 factors related to topography, geology, and hydrology, a normalized critical rainfall factor was suggested with an effective cumulative precipitation and a maximum hourly rainfall intensity (Yu et al. 2013a). The model produces a good assessment of the probability of occurrence of debris flows in the study area, and can be used in other regions.

Consequently, this paper presents a recent study on establishing one comprehensive threshold for predicting debris-flow occurrence based on the rainfall records and a possible new indicator – pore pressure measured under the ground surface. The purposes of this paper are: (i) to propose a new method for establishing a comprehensive threshold for forecasting debris-flow occurrence; (ii) to introduce the application and improvement of the comprehensive threshold with a case study.

#### 2. Study area

Wenjiagou gully located at the north of Qingping town, Mianzhu city, Sichuan province, Southwest China, has a catchment area of 7.8 km<sup>2</sup> and a 5.2 km long main channel, as shown in Fig. 1. The elevation of this study area ranges from 300 m to 1,600 m above sea level, and the valley with slope inclinations between 30° and 70° has been deeply incised by the Mianyuan river. The average yearly temperature of about 16  $^{\circ}$ C, and the climate is mild semi-tropical and moist with abundant rainfall and four distinguishable seasons. Eighty percent of the rainfall is concentrated in three months from July to September.

Fig. 1. Location of Wenjiagou gully modified from Huang et al. (2013). The inset photograph of Wenjiagou gully at the left bottom was taken from the other side of Mianyuan River on August 10, 2008.

Before the Wenchuan earthquake on May12, 2008, the Wenjiagou catchment was covered by rich vegetation, and the channel was smooth and stable. At that time, few geological disasters occurred in this region. Therefore, many farmers settled down at the foothills along the Mianyuan River.

After the earthquake, a giant landslide occurred upstream in the Wenjiagou catchment at the top of the watershed, which generated abundant co-seismic rock fall material and finer landslide deposits on a platform with an elevation of 1,300 m above sea level (Fig. 1, the photograph at left bottom of the main map). These loose solid erodible materials could transform into debris-flows during rainy season (Shieh et al. 2009). The catastrophic debris-flow triggered by a heavy rainfall on August 13, 2010, with a peak discharge of 1,530 m<sup>3</sup>/s and a total volume of  $4.5 \times 10^6$  m<sup>3</sup>, caused many victims and the