# Peer review of "Xiaoqiang XUE, Jian HUANG State Key Laboratory of Geohazard Prevention and Geoenvironment Protection Chengdu University of Technology, Chengdu, Sichuan 610059, China E-mail: huangjian2010@gmail.com"

_Natural Hazards and Earth System Sciences, 2016_

## Referee Comment (RC1) · Anonymous Referee #1 · 27 May 2016

I have read and evaluated the manuscript "A preliminary study on the comprehensive threshold for debris-flow early warning". Unfortunately, I did not find it suitable for publication in Natural Hazard and Heart System Sciences.

Below, some comments that may help the authors to improve their work before submitting it to other journals, more suitable to publish preliminary works like this one.

- Please, number the lines.

- English needs to be improved

- It is not clear to me what "comprehensive" means when it is referred to a rainfall threshold.

- The approach is too simple to be published in a top-rank journal as NHESS. It is very

simple with respect to the current state of the art of both empirical rainfall thresholds and physically based approaches.

- The introduction is very focused on China, discouraging the interest of a possible international audience.

- FIG 7: the way the line is drawn is very subjective and it is not supported by evidence. A threshold like I = 0.17R + 20 could be valid as well. Or even better, from a purely graphic point of view.

- In the manuscript there is a threshold linking I and E (rainfall parameters) and a series of threshold values of U. It is not clear to me how these two different aspects (ranifall paramenters and U) are linked together in the analysis and in the procedure of forecasting. They appear to stay disjointed.

- The test is represented by a SINGLE event that DID NOT trigger debris flows. This is not a good test: it does not support anything. Moreover, the threshold was exceeded (fig. 8).

---

## Referee Comment (RC2) · Anonymous Referee #2 · 5 Jun 2016

I have read the manuscript and I do not find it suitable for NHESS journal.

Generally the idea of threshold is interesting, but research lacks sufficient amount of data to determine the threshold. Authors present one case for determining threshold and one case for the test. What does comprehensive threshold stand for, when one is talking about precipitation?

Authors should get more data (more debris flow cases with measurements) to be able to perform analysis for threshold determination.

The idea in the manuscript is a good background for further research. Authors should consider developing methods which is not that China related and could be used on other testbeds.

---

## Author Comment (AC1) · 16 Jun 2016

A rainfall and pore pressure thresholds for debris-flow early warning: The Wenjiagou gully case study

X. Q. Xue & J. Huang huangjian2010@gmail.com Submitted on 29 Apr 2016, Received and Published: 10 May 2016

The authors would like to thank the two reviewers for their thorough work on the manuscript providing us with insightful and constructive comments and suggestions, which helped improve this manuscript. We have tried our best to carefully consider and respond to all the comments raised. The title of this manuscript has been changed to "A rainfall and pore pressure thresholds for debris-flow early warning: The Wenjiagou gully case study".

[Figure]

Response to Anonymous Referee #1 Thank the reviewer for the kindest article summary.

General Assessment: 1. Please, number the lines.

The revision has been added the numbers of lines.

2. English needs to be improved.

The language has been improved thoroughly and the text has been well edited.

3. It is not clear to me what "comprehensive" means when it is referred to a rainfall threshold. Compared to a single rainfall threshold, the pore pressure has been considered in this study. Therefore, the thresholds include these two factors.

4. The approach is too simple to be published in a top-rank journal as NHESS. It is very simple with respect to the current state of the art of both empirical rainfall thresholds and physically based approaches.

The method presented in this manuscript is simple, but it's still difficult to provide in these mountainous areas in Southwest of China particularly for the lack of valid data. However, the catastrophic disasters occurred every year in these regions, so that is so much urgent requirement for a simple and useful warning threshold under a limited information background.

5. The introduction is very focused on China, discouraging the interest of a possible international audience.

More information about the warning thresholds for debris flow presented by international researchers have been added in the revision.

6. FIG 7: the way the line is drawn is very subjective and it is not supported by evidence. A threshold like I = 0.17R + 20 could be valid as well. Or even better, from a purely graphic point of view. The threshold line is defined by a probabilistic and empirical method presented by Zhuang et al. (2014) and Huang et al. (2015b). This

method has been used directly in this work for lack of enough data and urgent demand. Zhuang J-q, Iqbal J, Peng J-b, Liu T-m (2014) Probability prediction model for landslide occurrences in Xi'an, Shaanxi Province, China Journal of Mountain Science 11:345-359. Huang J, Ju NP, Liao YJ, Liu DD (2015b) Determination of rainfall thresholds for shallow landslides by a probabilistic and empirical method Natural Hazards and Earth System Sciences 15:2715-2723 doi:10.5194/nhess-15-2715-2015.

7. In the manuscript there is a threshold linking I and E (rainfall parameters) and a series of threshold values of U. It is not clear to me how these two different aspects (rainfall parameters and U) are linked together in the analysis and in the procedure of forecasting. They appear to stay disjointed. This manuscript has been greatly improved to express itself in a clear way. During this study, rainfall and pore pressure are both considered in the warning threshold. First, the relationship between the two factors are analysed. Then, a threshold combined with both of them are presented.

8. The test is represented by a SINGLE event that DID NOT trigger debris flows. This is not a good test: it does not support anything. Moreover, the threshold was exceeded (fig. 8).

During this work, one single event was introduced in this manuscript for a better explanation about the approach. Surely, it can be seen that the rainfall threshold was exceeded, but the pore pressure was not exceeded during the rainfall event. Therefore, there is no debris-flow occurred during this rain event. Finally, this case study shows that the presented threshold is a useful tool for debris-flow prevention and mitigation in mountainous area at a preliminary stage. And the warning threshold can be improved and modified as long as more data are available during subsequent studies in the future.

Please also note the supplement to this comment:
http://www.nat-hazards-earth-syst-sci-discuss.net/nhess-2016-149/nhess-2016-149-AC1-supplement.pdf

[Figure]

**Supplement:**

**A rainfall and pore pressure thresholds for debris-flow early warning:**
**The Wenjiagou gully case study**

[revised manuscript text omitted]
^3$, caused many victims and the burying of houses, and the most downstream dam in the catchment (Yu et al. 2012).

**3.   Methodology**

According to Terzaghi theory in soil mechanics, the shear strength of material at a point within a slope can be expressed as Eq. (1).

$$t = c + (\sigma - \mu) \tan \phi \qquad (1)$$

where $t$ is the shear strength of the slope material, $c$ is the effective cohesion of the material,

$\phi$ is the effective friction angle of the material, $\sigma$ is the total stress normal to a potential slip surface, and $\mu$ is the pore-water pressure. Generally, the strength parameters ($c, \phi$) of the slope material mainly determined the stability of the slope and the potential position of the slip surface, also including the parameter ($\sigma$) determined by the height and inclination of the slope and the density of the slope material, and the distribution of pore-water pressure ($\mu$) within the slope.

Rainfall infiltrates into a hillslope, always accumulating in a saturated zone above a permeability barrier, and increases the pore-water pressures within the slope material. Based on the Terzaghi's work, the increase in $\mu$ would cause the effective overburden stress ($\sigma - \mu$), and therefore combining with the decrease of the shear strength until the slope fails. Presumed by Keefer et al. (1987), there exists a critical level of the pore-water pressure ($\mu_c$) for any given slope, facilitating the potential slip surface to develop, and causing the slope become unstable. In order to propose a formula to calculate the critical level of the pore-water pressure, a highly idealized model of an infinite slope composed of cohesionless materials ($c = 0$) has been presented in Keefer et al. (1987), where both slip surface and piezometric surface are parallel to the ground surface. For all these assumptions, the critical pore-water pressure can be calculated by Eq. (2).

$$\mu_c = Z \times \gamma_t \times \left( 1 - \frac{\tan \theta}{\tan \phi} \right) \qquad (2)$$

where $Z$ is the depth of slip surface, $\gamma_t$ is the total unit weight of the slope material, and $\theta$ is the slope inclination, the other parameters are the same to those mentioned-above.

In this respect, the Wenjiagou gully was selected as the case study area, and the pore-water pressure and rainfall monitoring sensors were installed in-situ to capture the real-time data for verifying these formulas and put forward a warning threshold for forecasting debris-flow occurrence finally. For this purpose, the history events about rainfall with debris-flow occurrences and non-occurrences have been collected during the initial research, unfortunately, none of those data has any pore-water pressure information. Even though several years have already passed under the real-time monitoring system, there are still limited available data for this research (Huang et al.

2015a).

**3.1 Data analysis**

Several methods were provided to collect available data, including debris-flow inventory maps, technical reports and documents presented by government agency. Since there is a large difference in debris flow frequency before and after the Wenchuan earthquake, only the data after quake were used for the analyses (Table 1).

Table 1 shows that the number of debris-flows decreases with time. Two years after the earthquake, however, several giant debris-flows still caused catastrophic losses, which alarmed the public and government because of its huge destructive power and long-term impact. Particularly on Aug. 13,

2010, a great rainstorm lasting for 2 hours during midnight, triggered a giant debris flow, which buried the Qingping town in the Mianyuan River floodplain. According to the inventory report, the maximum deposition height was until 6 m. Most of the check dams located in the downstream part of the

Wenjiagou gully collapsed and lost their effectiveness after passing of the debris-flow. It eroded the channel bottom over a depth of about 13 m (Yu et al. 2012).

Table 1. Primary rainfall events in Wenjiagou gully (2008-2011), from Xu (2010) & Yu et al. (2012)

| Time | Maximum hourly rainfall intensity ($I_h$: mm/h) | Accumulated precipitation ($R_{dt}$: mm) | Debris-flow occurrence or not | Volume of debris flow ($m^3$) |
|---|---|---|---|---|
| Sep. 24, 2008 | 30.5 | 88.0 | Yes | $5.0 \times 10^5$ |
| Jul. 18, 2009 | 20.5 | 70.5 | No | - |
| Aug. 25, 2009 | 28.9 | 86.7 | No | - |
| Sep. 13, 2009 | 15.4 | 84.6 | No | - |
| May 27, 2010 | 10.5 | 34.9 | No | - |
| Jun. 13, 2010 | 5.5 | 95.1 | No | - |
| Jul. 25, 2010 | 11.6 | 89.6 | No | - |
| Jul. 31, 2010 | 51.7 | 60.2 | Yes | $1.0 \sim 2.0 \times 10^5$ |
| **Aug. 13, 2010** | **70.6** | **227.0** | **Yes** | **$4.5 \times 10^6$** |
| Aug. 19, 2010 | 31.9 | 72.6 | Yes | $3.0 \times 10^5$ |
| Sep. 18, 2010 | 29.0 | 52.0 | Yes | $1.7 \times 10^5$ |
| Sep. 22, 2010 | 24.5 | 81.2 | No | - |
| May 2, 2011 | 5.6 | 35.8 | No | - |
| Jul. 5, 2011 | 12.5 | 61.3 | No | - |
| Jul. 21, 2011 | 23.5 | 63.2 | No | - |
| Jul. 30, 2011 | 18.2 | 78.3 | No | - |
| Aug. 16, 2011 | 10.5 | 44.3 | No | - |
| Aug. 21, 2011 | 13.6 | 76.6 | No | - |
| Sep. 7, 2011 | 15.2 | 51.3 | No | - |
| Oct. 27, 2011 | 8.5 | 36.9 | No | - |

Therefore, pore-water pressure and rainfall monitoring sensors have been installed for the relationship analyzes between rainfall, pore-water pressure and debris-flow occurrence. The real-time monitoring system in the Wenjiagou gully includes 7 automatic rain gauges and 5 pore-water pressure monitoring instruments, which have been completely finished until April 1, 2012, as shown in Table 2

and Figure 2. It can be seen that all rain gauges are arranged at the upstream of each branch gully in the Wenjiagou gully, and pore pressure sensors are located along the mainstream of the Wenjiagou gully in the deposits.

In 2012, a heavy rainfall event on August 14 triggered a debris-flow, which has been caught totally by the real-time monitoring system. During the rainstorm, monitoring sensors YL05, YL06 and SY02,

SY05 lost the connection with the monitoring center. The other monitoring sensors worked well, as shown in Figure 2, Figure 3 and Figure 4.

                          Table 2. List of monitoring devices in Wenjiagou gully

| No. | Longitude | Latitude | Elevation(m) | Purposes |
|---|---|---|---|---|
| YL01 | E104°8'21" | N31°33'32" | 1652 | To gain the rainfall in the study area, and analyze the relationship between rainfall and debris-flow occurrence. |
| YL02 | E104°7'55" | N31°33'11" | 1390 | |
| YL03 | E104°8'39" | N31°33'14" | 1671 | |
| YL04 | E104°8'16" | N31°32'47" | 1490 | |
| YL05 | E104°7'47" | N31°32'39" | 1433 | |
| YL06 | E104°7'46" | N31°33'29" | 1166 | |
| YL07 | E104°7'9" | N31°32'59" | 1025 | |
| SY01 | E104°8'12" | N31°33'9" | 1210 | To gain the pore-water pressure, and analyze the relationship between pore-water pressure and debris-flow occurrence. All of them were buried at a depth of 1 m under the ground. |
| SY02 | E104°8'11" | N31°33'9" | 1212 | |
| SY03 | E104°8'11" | N31°33'8" | 1208 | |
| SY04 | E104°7'49" | N31°32'55" | 1092 | |
| SY05 | E104°7'48" | N31°32'56" | 1081 | |

[Figure]

Fig. 2. Layout map of the monitoring devices installed in Wenjiagou gully ( The base map is from Google Earth,
                the date of background image is Dec. 18, 2010).

Figure 3 and Figure 4, show that the rainfall was almost concentrated in two hours from 17:00

until 19:00. The amount of precipitation was highly variable along the channel of the Wengjiagou gully. The maximum hourly rainfall intensity is 73.5 mm (YL01), and the standard deviation is 12.76.

The cumulative maximum rainfall is 118 mm (YL04), and the standard deviation is 14.75.

[Figure]

Fig. 3. The rainfall in Wenjiagou gully on Aug. 14, 2012 (the column maps are hourly rainfall and the single line

   maps are cumulative rainfall)

[Figure]

Fig. 4. The rainfall and pore-water pressure in Wenjiagou gully on Aug. 14, 2012 (the column maps are hourly

   rainfall and the single line maps are pore-water pressure)

The maximum hourly rainfall and cumulative rainfall are not found in the highest part of the catchment. The variety in cumulative maximum rainfall is larger than the variety in maximum hourly rainfall intensity. The Figure 4 shows the relation between hourly rainfall and pore-water pressure: the small amount of rain from 2:00 to 5:00 with a maximum hourly rainfall of 12.5 mm did not trigger any change in pore-water pressure. However, during the concentrated rain period between 17:00 and 19:00

there was a sudden rise of the pore-water pressure. The debris flow was triggered adjacently when it reached the maximum rise of the pore-water pressure. The highest value of the pore-water pressure are

9.01 kPa (SY01) at 17:00, 5.7 kPa (SY03) at 20:00 and 4.17 kPa (SY04) at 18:00. The sudden rise of pore-water pressure may be a good indicator for the triggering of debris-flows.

**3.2 Warning threshold for Wenjiagou gully**

In order to improve the warning thresholds for forecasting the debris flow occurrence, which not just represent a simple relationship between rainfall and debris-flow occurrence, the pore-water pressure of landslide deposits was incorporated into the threshold for it is an important geotechnical parameter in the physical models of debris-flow generation. Back to the critical pore-water pressure

Eq.(2) above-mentioned, combing with the results from field investigation in the Wenjiagou gully on

Oct. 8, 2010, the total unit weight of landslide deposits in the Wenjiagou gull is 21.05 kN/m$^3$, the slope inclination of 18.5°, and the effective friction angle of the deposited material is 27.5°. Thus, the critical pore-water pressure of the deposited material can be calculated by Eq. (3).

$$\mu_c = 7.52 \times Z \tag{3}$$

It's a linear function, which can be shown in a graph (Figure 5). If the slip surface of the deposited material is at the position of 1 m depth under the ground, the critical pore-water pressure should be

7.52 kPa. With a redundant consideration of 25% in depth of slip surface, the critical pore pressure is between 5.64 kPa and 9.40 kPa (Figure 5). However, up to present in the real-time monitoring system, only 3 events of debris-flow occurrences were captured (August 14, 2012; August 17, 2012 and July 9,

2013). When the debris flow occurred, the pore-water pressure and cumulative rainfall information are demonstrated with line graph and column graph respectively (Figure 6). The average of monitoring critical pore-water pressure decrease with time, comparing to the rise of cumulative rainfall. It indicates that the critical pore-water pressure for debris-flow occurrence can be revised and adjusted

190 by itself. Meanwhile, the theoretic critical pore-water pressure ($\mu_c = 7.52\ \text{kPa}$) can be used to

191 predict the debris-flow occurrence, by considering a probability of 25% in the depth of slip surface.

192 Therefore, the thresholds of critical pore-water pressure from 5.64 kPa to 9.40 kPa can be used to

193 supplement the warning threshold for debris-flow prediction in the Wenjiagou gully.

[Figure]

[Figure]

Fig. 5. The theoretic critical pore-water pressure of  Fig. 6. The monitoring critical pore-water pressure of
deposited material in the Wenjiagou gully    deposited material in the Wenjiagou gully

194  Considering the acquired available data, based on the methodology above methioned, the

195 maximum hourly rainfall ($I_h$: mm) and cumulative rainfall ($R_t$: mm) are selected as the basic triggering

196 rainfall parameters for the warning threshold, and the theoretical critical pore-water pressure ($U_c$) has

197 been defined as an assistant factor in forecasting debris-flow occurrence. For each rainfall event with

198 or without debris-flow occurrence, $R_t$ and $I_h$, can be plotted in a X-Y field, like the debris-flow event

199 on Aug. 13, 2010 (Fig. 7 Tag A). The rainfall threshold for predicting debris-flow occurrence can be

200 defined and calculated by Eq. (4) and Figure 7 (Tag C). The line separates these debris-flow

201 occurrences from non-occurrences, and indicating that when rainfall starting the point (cumulative

202 rainfall & hourly rainfall intensity) can be calculated in a real-time and plotted into the diagram. While

203 the point crosses the line, which shows the probability of debris-flow occurrence is much higher. More

204 detail information can be found in Zhuang et al. (2014) and Huang et al. (2015b).

205         $$R_t + 2.4I_h = 120 \tag{4}$$

206 where $R_t$ is the cumulative rainfall (mm), $I_h$ is the maximum hourly rainfall (mm).

[Figure]

  Fig. 7. Rainfall threshold based on maximum hourly rainfall and cumulative rainfall

Figure 7 shows thirteen points (86.7%) of collected rainfall events (Table 1) with no debris flows lying below the threshold line, and two error points above this line. However, as mentioned above, only the rainfall threshold may not be enough to predict debris-flow occurrence. Pore-water pressure, therefore, has been selected to refine the warning threshold. Figure 5 and Figure 6 show that the critical pore-water pressure ($U_c$) above a certain threshold is also a valuable indicator for forecasting debris-flow occurrence. Based on the verification of theoretical and real-time monitoring pore-water pressure, the threshold for pore-water pressure (U) to predict the probability of debris flow occurrence, can be defined by Eq. (5) based on the Eq. (2) and Eq. (3).

                               $$U \geq U_c \qquad\qquad\qquad (5)$$

When both of the two thresholds are satisfied (Eq.4 & Eq.5), there must be a very high possibility of debris-flow occurrence. Generally, more available data will improve the warning thresholds, and make them more reliable and accurate for debris-flow prediction.

**4.  Example of application**

In order to make a better use for the debris-flow early warning, some criteria have to be simplified

223 for the preliminary stage. Therefore, a three-level early warning system has been proposed for the

224 Wenjiagou gully, as shown in Table 3.

225 Table 3. Recommended warning levels for Wenjiagou gully

| Warning level | Trigger | Response |
|---|---|---|
| I | Default level. Eq. (4) is not satisfied or U ≤ 5.64 kPa. | **Null**: but data are checked daily. Weekly monitoring bulletin. |
| II | Attention level. Eq. (4) is satisfied or 5.64 kPa < U < 9.40 kPa. | **Watch**: data are checked more frequently. Daily monitoring bulletin. Authority and expert are alerted. Preparing for alarm. |
| III | Alert level. Eq. (4) is satisfied or U ⩾ 9.40 kPa. | **Warning**: data are checked even more frequently. Two monitoring bulletins per day. Local people are alerted. |

226 At level one (Blue) there is no risk of debris-flows. At level two (Orange) there is a chance of

227 debris-flow occurrence in the near future, and warning messages need to be sent to local authority and

228 countermeasures need to be discussed. Level three (Red) there is very likely to occur right now,

229 therefore, local residents need to be alerted and forbidden going to that place.

230 In order to show how this presented method can be used in debris-flow early warning, a heavy

231 rainfall event on Jun. 19, 2013 (YL01 & SY01) has been selected as a case application (Figure 8). The

232 red circle solid points give the real-time monitoring rainfall information of the event, with cumulative

233 rainfall on the X axis and hourly rainfall intensity on Y axis. The Tag A in Figure 8 shows the

234 monitored rainfall at 7:00 am on Jun. 19, 2013, and the U-value of the pore-water pressure is 7.00 kPa

235 at that time. One hour later at 8:00 am (Tag B), the real-time rainfall has exceeded the rainfall

236 threshold, but the maximum measured U-value didn't met the critical value given in Eq. (5) (5.64 kPa

237 < U=7.60 kPa < 9.40 kPa) which indicated the warning level stayed in Orange. Finally, the rainfall

238 went down after 8:00 am, and no debris-flow occurred during this rain event.

[Figure]

Fig. 8. Case application of the presented method in Wenjiagou gully (Jun. 19, 2013)

This case study shows, that it is worthwhile to continue to test the presented warning threshold as a useful tool for debris-flow prevention and mitigation in mountainous area at a preliminary stage. The alert threshold can be improved and modified as long as more data are available during subsequent studies in the future.

**5.    Discussion and conclusion**

Debris-flow, usually triggered by rainstorm every year in the mountainous region in Southwest

China, always cause significant harm both in human and property losses. Therefore, in order to prevent such natural disaster there is an urgent requirement for effective methods to predict debris-flow occurrence. The warning thresholds are provided and discussed in this paper, which not only use the common rainfall threshold, but also include the critical pore-water pressure from theoretical analysis of slope stability.

Two triggering factors: maximum hourly rainfall versus cumulative rainfall, and pore-water pressure, have been selected to establish the warning thresholds for debris-flow occurrence prediction.

The Wenjiagou gully was selected as a case study for a detailed explanation of the provided method, and the results show that it is worth the effort to test further this approach for the early warning of debris-flows, especially at the preliminary stage. However, the assessment of such a warning threshold cannot be extended, to larger areas. Moreover the collection of pore-water pressure data in a distributed way is a rather cumbersome and costly undertaking and will be restricted to dangerous catchments like the Wenjiagou gully. Another the complicated problem is the final determination whether to alert local population, and whether compulsory actions need to be done at once, or a period of time later. Debris-flow early warning is not an imminent hazard but is just regarded as a potential danger. In spite of these limitations, the method has reached the goal to establish a warning threshold for debris-flow early warning, and more subsequent work will be carried on in the future.

**Acknowledgements**

This study was financially supported by the National Natural Science Foundation of China (Grant

No. 41521002 and No. 41302242), Specialized Research Fund for the Doctoral Program of Higher

Education of China (Grant No. 20135122130002), and State Key Laboratory of Geo-hazard

Prevention and Geo-environment Protection (Chengdu University of Technology) (Grant No.

SKLGP2013Z007). The authors also give great thanks to Prof. Theo van Asch for his review of an earlier version of this paper and for his suggestions to polish the language, which greatly improved the quality of the manuscript.